

# Comparative analysis of machine learning approaches to analyze and predict the COVID-19 outbreak

Muhammad Naeem[1], Jian Yu[2], Muhammad Aamir[1], Sajjad Ahmad Khan[3], Olayinka Adeleye[2] and Zardad Khan[1]

[1] Department of Statistics, Abdul Wali Khan University, Mardan, KP, Pakistan
[2] Department of Computer Science, Auckland University of Technology, Auckland, New Zealand
[3] Department of Statistics, Islamia College University, Peshawar, KP, Pakistan

## ABSTRACT

**Background:** Forecasting the time of forthcoming pandemic reduces the impact of diseases by taking precautionary steps such as public health messaging and raising the consciousness of doctors. With the continuous and rapid increase in the cumulative incidence of COVID-19, statistical and outbreak prediction models including various machine learning (ML) models are being used by the research community to track and predict the trend of the epidemic, and also in developing appropriate strategies to combat and manage its spread.

**Methods:** In this paper, we present a comparative analysis of various ML approaches including Support Vector Machine, Random Forest, K-Nearest Neighbor and Artificial Neural Network in predicting the COVID-19 outbreak in the epidemiological domain. We first apply the autoregressive distributed lag (ARDL) method to identify and model the short and long-run relationships of the time-series COVID-19 datasets. That is, we determine the lags between a response variable and its respective explanatory time series variables as independent variables. Then, the resulting significant variables concerning their lags are used in the regression model selected by the ARDL for predicting and forecasting the trend of the epidemic.

**Results:** Statistical measures—Root Mean Square Error (RMSE), Mean Absolute Error (MAE), Mean Absolute Percentage Error (MAPE) and Symmetric Mean Absolute Percentage Error (SMAPE)—are used for model accuracy. The values of MAPE for the best-selected models for confirmed, recovered and deaths cases are 0.003, 0.006 and 0.115, respectively, which falls under the category of highly accurate forecasts. In addition, we computed 15 days ahead forecast for the daily deaths, recovered, and confirm patients and the cases fluctuated across time in all aspects. Besides, the results reveal the advantages of ML algorithms for supporting the decision-making of evolving short-term policies.

# INTRODUCTION

The outbreak of the novel coronavirus disease in 2019 (COVID-19) has emerged as one of the most devastating respiratory diseases since the 1918 HIN1 influenzas pandemic, infecting millions of people globally (*Tuli et al., 2020*). The cumulative incidence of the

Corresponding author
Muhammad Aamir,
aamirkhan@awkum.edu.pk

virus is continually and rapidly increasing globally. At the early stage of the outbreak, it is important to have a clear understanding of the disease transmission and its dynamic progression, so that relevant agencies and organizations can make informed decisions and enforce appropriate control measures. Generally, capturing the transmission dynamics of a disease over time can provide insights into its progression, and show whether the outbreak control measures are effective and able to reduce the impact of the disease on a community (*Kucharski et al., 2020*).

Access to real-time data and effective application of outbreak prediction or forecasting models are central to obtaining insightful information regarding the transmission dynamics of the disease and its consequences. Moreover, every outbreak has its unique transmission characteristics that are different from the other outbreaks, which raises the question of how standards prediction models would perform in delivering accurate results. In addition, various factors including the number of known and unknown variables, differences in population/behavioural complexity in various geopolitical areas, and the variations in containment strategies increase the uncertainty of prediction models (*Ardabili et al., 2020*). As a result, it is challenging for standard epidemiological models such as Susceptible-Infected-Recovered (SIR) to provide reliable results for long-term predictions. Therefore, it is important to not only study the relationship between the components of the outbreak datasets but also evaluate the effectiveness of the common disease prediction models.

In recent months, there have been a handful of works that try to understand the spread of COVID-19, especially using statistical approaches. For instance, *Kucharski et al. (2020)* explored a combination of stochastic transmission model and four datasets that captured the daily number of new cases, the daily number of new internationally exported cases, the proportion of infected passengers on evacuation flight and the number of new confirmed cases, to estimate the transmission dynamics of the disease over some time.

In another study, a machine learning-based model is applied to analyse and predict the growth of COVID-19 (*Tuli et al., 2020*). The authors demonstrated the effectiveness of using iterative weighting for fitting Generalized Inverse Weibull distribution when developing a prediction solution. *Lin et al. (2020)* presented a conceptual model designed for the COVID-19 epidemic with consideration of individual behavioural responses and engagements with the government, including the extension in holidays, restriction on travel, quarantine, and hospitalization. This work combined zoonotic transmission with the emigration pattern, and then estimate the future trends and the reporting proportion. The model gives promising insight into the trend of the COVID-19 outbreak, especially the impact of individual and government reactions or responses to the epidemic. *Anastassopoulou et al. (2020)* estimated the average values of the key epidemiological parameters including the per day case mortality, recovery ratios, and the basic reproduction number $R_0$ representing the average number of ancillary cases that results from the introduction of a single infectious case in an entirely susceptible population during the active period of the pandemic. The authors fit the dataset to the Susceptible-Infectious-Recovered-Dead (SIDR) model and attempted a 3-week prediction of the dynamics of the outbreak at the epic centre. The estimated mean value of $R_0$ as calculated

considering the period from the 11th January to the 18th of January was found to be around 2.6 based on the official confirmed cases. *Hu et al. (2020)* proposed a machine learning approach to predict the magnitude, intervals, and completion period of the disease. The authors proposed an improved auto-encoder model to analyse the spread-changing aspects of the epidemics then predict the definite cases. In the model, hidden variables are used to group the cities for probing the spread arrangement. By means of the many-step predicting, the expected errors of 6, 7, 8, 9 and 10-step predicting remained 1.64%, 2.27%, 2.14%, 2.08%, 0.73%, correspondingly.

Autoregressive Distributed Lag (ARDL) is a flexible method to include independent series in dynamic regression models. ARDL models contain previous values of together response and explanatory variables series. They have been widely used in various domains including marketing, energy, epidemiology, agronomy, and ecological studies (*Huffaker & Fearne, 2019*). Over the years, many packages have been developed for ARDL; for example, see *Pesaran, Shin & Smith (2001)*. The distributed lag model has a wide range of applications that is cointegration study in which small and large-run relations between time series data. ARDL boundaries testing of *Pesaran, Shin & Smith (2001)*, which is a common co-integration study technique founded on the distributive lag model and further research work in progress.

The other package developed by *Demirhan (2020)* is `nardl` to use Distributed Lag Models (DLMs) in R. The package `nardl` focuses on the application of the nonlinear cointegrating ARDL model is developed by *Shin, Yu & Greenwood-Nimmo (2014)*. The recent package `dynlm` takes a unique purpose to fit linear models *via* stabilizing time-series features (*Zeileis & Zeileis, 2019*).

*Satu et al. (2021)* used the machine learning models to predict the COVID-19 cases for the short-term prediction. The authors used the regression-based ML models for short-term forecasting. In the same way, *Mojjada et al. (2020)* used machine learning methods and it is capable to predict the COVID-19 affected persons. The authors used four different types of ML approaches namely, the Lower Absolute Reductor and Selection Operator (LASSO), exponential smoothing (ES), Linear Regression (LR) and Support Vector Machine (SVM). These models is used to predict the number of deaths, recovered and newly infected COVID-19 individuals for the next 10 days. In his studies, the linear model is most effective in predicting these cases.

In this work, we present a comparative analysis of various machine learning approaches including Support Vector Machine (SVM), Random Forest (RF), K-Nearest Neighbor (KNN) and Artificial Neural Network in predicting the COVID-19 outbreak in the epidemiological domain. We aim to determine how well each of these approaches performs in predicting the confirmed and death cases and then compare their performances with each other. Particularly, we first apply the ARDL method to identify and model the short and the long-run relationships of the time-series COVID-19 datasets (confirmed, recovered and death cases). That is, we determine the lags between a response variable and its respective explanatory time series variables as independent

variables. Then, the resulting significant variables concerning their lags are used in the regression model selected by the ARDL model for predicting and forecasting the trend and dynamics of the COVID-19. We evaluated the models using relevant accuracy and error metrics including Root Mean Square Error (RMSE), Mean Absolute Error (MAE), Mean Absolute Percentage Error (MAPE) and Symmetric Mean Absolute Percentage Error (SMAPE).

## MATERIALS AND METHODS

### Data source

We conducted our study based on the publicly accessible data of daily deaths, recovered and confirmed cases 671127, 10585 and 309869 respectively reported for all over the world from 22nd January 2020 to 18th August 2021 in Fig. 1. The data is available in the online repository-GitHub (https://github.com/CSSEGISandData/COVID-19).
We perform data processing including the conversion of data format from cumulative to daily basis. This repository is for the COVID-19 visual dashboard operated by Johns Hopkins University Centre Systems Science and Engineering (JHU CSSE). They have aggregated data from sources like WHO, WorldoMeters, BNO News, and Washington State Department of Health and many more. The data have the number of confirmed cases, the recovered cases, and the death cases for the globe. On this data, we attempted to forecast the key epidemiological parameters, that is, the number of upcoming daily new confirmed cases, deaths, and recoveries. Though, the quantity of deaths, recovery and confirmed cases of individuals is expected to be much higher over time. Therefore, we have similarly derived a correlation between these two variables and their past record (lags) by using the ARDL model.

### Autoregressive distributive lag models

The ARDL models are used between regressed series and $k$ number of regressors series in regression analysis. If there is only one independent series, the dependent lag series makes the model autoregressive. The numeral of $p^{th}$ independent lag series is denoted by $p_j$, $j = 1, \ldots, n$ denotes daily recovery and confirm cases, the $q^{th}$ lags of dependent variable series are shown by $q_i$, where $i = 0, 1, \ldots, m$.

The ARDL model can be expressed as:

$$y_t = \alpha_0 + \beta_1 y_{t-1} + \beta_2 y_{t-2} + \ldots, \; \beta_j y_{t-i} + \gamma_1 x_t + \gamma_2 x_{t-1}$$
$$+ \ldots, \gamma_i x_{t-j} + \delta_1 w_t + \delta_2 w_{t-1} + \ldots, \; \delta_i w_{t-j} + \; \varepsilon_t \tag{1}$$

where $y_t$ denotes the number of daily deaths at time $t$. $\alpha_0$ represent the intercept term. In the same way, $\beta_1 y_{t-1} + \beta_2 y_{t-2} + \ldots, \; \beta_q y_{t-i}$ denotes the $q^{th}$ autoregressive lag order of the model of the dependent variable. The two independent variables "recover cases" and "confirm cases" are denoted by $x_t$ and $w_t$ respectively. Whereas $\gamma_1 x_t + \gamma_2 x_{t-1} + \ldots,$ $\gamma_i x_{t-j}$ and $\delta_1 w_t + \delta_2 w_{t-1} + \ldots, \; \delta_i w_{t-j}$ represent the lags order of $x_t$ and $w_t$ respectively. The parameters $\beta$, $\gamma$ and $\delta$ denoted coefficients of death, recovery, and confirmed cases,

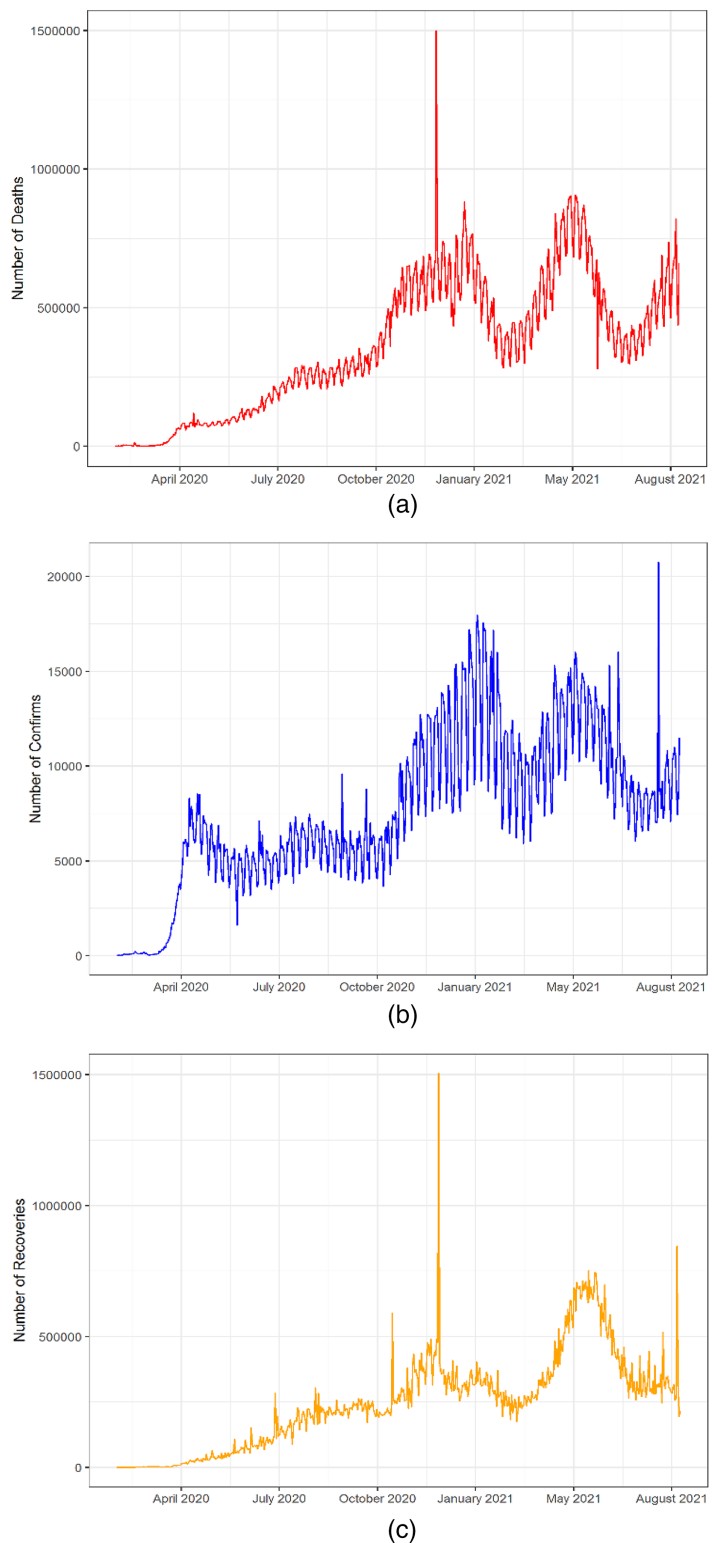

**Figure 1 Plots of the daily deaths, recovered and confirmed COVID-19 outbreak.** (A) Confirmed (Minimum = 380, Maximum = 1,498,044) (B) Recoveries (Minimum = 9, Maximum = 1,504,943) & (C) Deaths cases (Minimum = 2, Maximum = 20,752).

respectively, while $\varepsilon_t$ denotes the error term. Eq. (1) can be further simplified and presented in Eq. (2):

$$y_t = \alpha_0 + \sum_{i=1}^{m} \beta_i y_{t-i} + \sum_{j=0}^{n} \gamma_j x_{t-j} + \sum_{j=0}^{n} \delta_j w_{t-j} + \varepsilon_t \tag{2}$$

The number of deaths, confirm, and recover cases of people is likely to be much higher with time. Therefore, the ARDL model for recovered cases $x_t$ and confirmed cases $w_t$ is shown in Eq. (3).

$$x_t = \theta_0 + \gamma_1 x_{t-1} + \ldots, \gamma_i x_{t-j} + \delta_1 w_t + \delta_2 w_{t-1} + \ldots, \delta_i w_{t-j} + \varepsilon_t \tag{3}$$

Similarly, the ARDL model for confirmed and recovered cases is shown in Eq. (4)

$$w_t = \vartheta_0 + \delta_1 w_t + \delta_2 w_{t-1} + \ldots, \delta_i w_{t-j} + \gamma_1 x_{t-1} + \ldots, \gamma_i x_{t-j} + \varepsilon_t \tag{4}$$

There are different criteria used to select an optimal lag length selection. The authors in *Chandio, Jiang & Rehman (2020)* use Akaike Information Criterion (AIC) and the authors in *Gayawan & Ipinyomi (2009)* compare AIC, *SIC* and adj-R square to select the optimal lag length. We use adj-R square and parsimony model criteria to select an optimal number of the lag length in this study. It makes the call to the function easier when the number of lags order are the same, however, when the number of lags order is different from dependent and every independent sequence, we use the argument *remove*. It will remove the lags that are not contributed to the model. Once the ARDL model specifies the significant coefficients of the dependent variable and independent variables, the models including the RF, SVM, KNN, and ANN are used to assess the accuracy and error rate of these models. We utilized RF (*Biau & Scornet, 2016*), SVM (*Liang et al., 2018*), KNN (*Martínez et al., 2019*) and ANN (*Hu et al., 2018*) time series models were applied to predict the COVID-19. To overcome the overfitting problem, we use 80% training and 20% testing parts, respectively. Random forest is one of the best learning algorithms and it requires a bit of parameter tuning.

Generally, in time series analysis, Support Vector Regression (SVR) is used. In SVM, various kernel functions are used to develop the input space into a feature space with a complex dimension. Like Gaussian Radial Basis (GRBF), Sigmoid, polynomial, *etc.* are some kernel functions. For SVM, we use Radial Basis Kernels (RBF) $k_\gamma(y_i, y_j) = \exp(-\gamma \|y_i - y_j\|)^2$. In the SVM model, using RBF kernels it is necessary to tune model parameters to find an optimal value of the parameters and reducing the overfitting problem. So, we use the grid search method of tenfold cross-validation on the training part and testing part and their results are averaged.

K-nearest neighbor (k-NN) predicts the response variable based on the nearest training points. It uses a training dataset in its place of learning a discriminative function from the training data. k-NN is used both for classification and regression problems. There are various techniques use to improve model accuracy. Such as maximum percentage accuracy

**Table 1 Forecasting evaluation measurement tools.**

| Criterion formula |
| --- |
| Mean error $ME = \frac{1}{n}\sum_{t=1}^{n}(Y_t - \hat{Y}_t)$ |
| Root mean square error $RMSE = \sqrt{\frac{1}{n}\sum_{t=1}^{n}(\hat{Y}_t - Y_t)^2}$ |
| Mean absolute error $MAE = \frac{1}{n}\sum_{t=1}^{n}|\hat{Y}_t - Y_t|$ |
| Mean percentage error $MPE = \frac{1}{n}\sum_{t=1}^{n}(\frac{\hat{Y}_t - Y_t}{Y_t})*100$ |
| Mean absolute percentage error $MAPE = \frac{1}{n}\sum_{t=1}^{n}|\frac{\hat{Y}_t - Y_t}{Y_t}|*100$ |
| Symmetric Mean absolute percentage error $SMAPE = \frac{1}{n}\sum_{t=1}^{n}|\frac{\hat{Y}_t - Y_t}{(Y_t + \hat{Y}_t)/2}|*100$ |
| R square $R^2 = 1 - \frac{\sum_{t=1}^{n}(Y_t - \hat{Y}_t)^2}{\sum_{t=1}^{n}(Y_t - \bar{Y})^2}$ |

graph, Elbow method, for loops to select an optimal value of k. Generally, the square root of n is used, and we utilized $\sqrt{n}$.

ANN is a mathematical tool and has been generally used for classification and forecasting problems properly that contain predictors (input) and response (output) layers, and a hidden layer. A combination of different hidden layers is used to choose a better MLP architecture network. It is the hidden layers in ANN models that play an important role in many successful applications of neural networks. ANN model is widely used in economic and financial studies (*Huang et al., 2007*; *Qi, 1996*). The number of hidden layers depends upon the nature of the problem. The authors (*Zhang, Patuwo & Hu, 1998*) used two hidden layers and finds better model prediction accuracy. In the same way, the authors in *Xu et al. (2020)* used $(2 \times k + 1)$, where $k$ is the number of predictors (inputs). For an optimal result of ANN, usually, trial and error method is used in determining the number of hidden nodes that is searching the architecture having the smallest MAPE among the models (*Güler & Übeyli, 2005*). We use four hidden layers and eight neurons in the hidden layers for daily death cases using trial and error procedure and 10,000 times iteration. In the same way, we use two hidden layers and four number of neurons in the hidden layers for daily recover cases.

## Forecast evaluation criterions

In this study, as the response variable is continuous, therefore, the forecasting capacity of different machine learning approaches are evaluated by using five different criteria including mean error (ME), RMSE, Mean Absolute Error (MAE), Mean Percentage Error (MPE), MAPE, SMAPE, and R square presented in Table 1. Where $n$ represents the total number of prediction on training and testing parts respectively, $Y_t$ and $\hat{Y}_t$ representing the observed and predicted values, respectively.

## Software and packages

In the current study, we will use the R programming version 4.0.4 and `dLagM` package that outfits the ARDL test method (*Pesaran, Shin & Smith, 2001*). Subsequently, `dLagM` uses lag orders, dataset, and overall method which make the prerequisite lags and changes for definite models. One of the benefits of this approach is that the users are not required to specify the variation for the applied models. Which brings efficacy and value to researchers in various areas.

In this study, we used `tseries`, `timeseries`, `zoo` and `window` packages for the data. In the same way, `dLagM` package in R for ARDL model. An orders $p$ and $q$ of the ARDL lag model are denoted by ARDL $(p, q)$, which has independent $p$ lags series and dependent $q$ lags series. We use the packages, `randomForest`, `forecast`, `caret`, `tiyverse`, `tsibble` and `purr` for RF. The `ntree` is 500, `mtry` is p/3, where p is the number of features, `sampsize` is 70% and `type` is "regression" utilized in the function. The other parameters are kept as default. In this study, the library `e1071` is used for SVM, the parameters $cost = 10^2$, $gamma(\gamma) = 0.1$, and the $insensitivity\ (\varepsilon) = 0.3$ respectively. In the same way, k-nearest neighbor Regression the `caret` package is used. For ANN, the `neuralnet` package is used. The parameters, the *algorithm, threshold*, and *linear.output* is 'backprop', 0.01, TRUE and the other parameters are kept as default, respectively.

## RESULTS

A total of three data sets of COVID-19 (confirm, recover and death) are used to evaluate the performance of the different ML approaches and suggested the best model for forecasting the COVID-19 outbreak. All data sets consisting of the world's daily confirm, recovery and death cases. Every time series divided into training and testing sets of observations. The original data divided into 80% training and 20% testing parts and the first 80% of the total observations in every time series used as a training set whereas the rest 20% used as the testing set. To overcome the overfitting problem, we use 10-fold cross-validation for each of the models and then their results are averaged. In addition, we also used prediction accuracy for training parts. Each time series containing a total of 574 observations spanning (22 January 2020, to 18 August 2021), the first 459 observations spanning (22 January 2020, to 24 April 2021) belong to the training series and the rest 115 observations spanning (25 April 2021, to 18 August 2021) part of the testing series.

We use death, recover, and confirm cases from the COVID-19 dataset. The COVID-19 dataset is loaded into the R package environment, and then, we fit the ARDL model to the Daily Deaths series $y_t$ with recover $R_t$ and confirm $C_t$ cases. We choose $p_1 = 3$, $p_2 = 3$, *and* $q = 2$ using R square and parsimony of the model. The insignificant variables are removed and fit the ARDL model. The results obtained from the ARDL model are presented in Table 2.

The coefficient related to confirm cases $C_t$ and its first lag is highly significant at 0.5% level, respectively. Similarly, the current death of the response variable $y_t$ (daily deaths of COVID-19), are significant at the 0.5% level. In addition, the coefficient of recover cases (current) are also significant at 0.5% level, respectively. Overall, the model is highly significant at the 0.5% level with a *p*-value smaller than $2.2 \times 10^{(-15)}$ with the R-squared

**Table 2 Summary of ARDL model 1 for daily deaths of COVID-19.**

| Coefficients | Estimate | Std. error | t value | P-value |
|---|---|---|---|---|
| (Intercept) | 1,905.14 | 5,434.08 | 0.350 | 0.7260 |
| Ct.1 | −0.176 | 0.036 | −4.77 | $2.34 \times 10^{(-06)}$*** |
| Ct.2 | 0.597 | 0.045 | 13.00 | $6.84 \times 10^{(-34)}$*** |
| Ct.3 | −0.262 | 0.035 | −7.31 | $9.18 \times 10^{(-13)}$*** |
| Rt.t | 8.93 | 1.14 | 7.79 | $3.21 \times 10^{(-14)}$*** |
| Rt.2 | −10.25 | 1.26 | −8.07 | $4.23 \times 10^{(-15)}$*** |
| Yt.1 | 0.340 | 0.037 | 9.05 | $2.38 \times 10^{(-18)}$*** |
| Yt.2 | 0.251 | 0.035 | 7.02 | $6.55 \times 10^{(-12)}$*** |

| | | | | |
|---|---|---|---|---|
| Residual standard error: 57,720 | | | | |
| R-squared: 0.9029 | | | Adjusted R-squared: 0.9014 | |
| F-statistic: 636.7 | | | P-value: $<2.2 \times 10^{(-14)}$ | |

**Note:**
*** Significant at 1%.

**Table 3 Summary of final ARDL model 2 for confirm and recover of COVID-19.**

| Coefficients | Estimate | Std. error | t value | P-value |
|---|---|---|---|---|
| (Intercept) | $7.96 \times 10^{(2)}$ | $1.59 \times 10^{(2)}$ | 5.00 | $7.6 \times 10^{(-07)}$*** |
| Ct.1 | $8.41 \times 10^{(-1)}$ | $4.20 \times 10^{(-2)}$ | 20.02 | $<2 \times 10^{(-16)}$*** |
| Ct.2 | $-2.21 \times 10^{(-1)}$ | $5.43 \times 10^{(-2)}$ | −4.07 | $5.2 \times 10^{(-05)}$*** |
| Ct.3 | $1.42 \times 10^{(-1)}$ | $4.12 \times 10^{(-2)}$ | 3.43 | 0.00059*** |
| Rt.1 | $4.25 \times 10^{(-3)}$ | $6.29 \times 10^{(-4)}$ | 6.75 | $3.5 \times 10^{(-11)}$*** |

| | | | | |
|---|---|---|---|---|
| Residual standard error: 1,727 | | | | |
| R-squared: 0.8294 | | | Adjusted R-squared: 0.8281 | |
| F-statistic: 670.8 | | | P-value: $2.1 \times 10^{(-15)}$ | |

**Note:**
*** Significant at 1%.

equal to 90.29% and the alpha value (*Benjamin et al., 2018*). The fitted model can be written as:

$$
\begin{aligned}
y_t(Daily\ Deaths) = {}& 1905.14 + 0.340y_{t-1} + 0.251y_{t-2} + 8.93x_t - 10.25x_{t-2} \\
& - 0.176w_{t-1} + 0.597w_{t-2} - 0.262w_{t-3} + \varepsilon_t
\end{aligned}
\tag{5}
$$

In the second scenario, we examine the relationship between the number of recover cases and confirm cases. We fit the ARDL model for recover cases $x_t$ of COVID-19 series with confirm $w_t$ cases. We take $p_1 = 4$, *and* $q = 3$ using R square and parsimony of the model and fitting the ARDL model to the datasets. The results obtained from the ARDL model are presented in Table 3.

Table 3 shows the summary of the ARDL model, the confirmed cases recorded in the current day. The daily recover cases of the first day have a significant impact on the number of daily confirm cases from the COVID-19 on that particular day. The model is

**Table 4 Summary of final ARDL model 3 for confirm and recover of COVID-19.**

| Coefficients | Estimate | Std. error | t value | P-value |
|---|---|---|---|---|
| (Intercept) | $-8.1 \times 10^{(3)}$ | $5.9 \times 10^{(3)}$ | $-1.37$ | 0.169 |
| Rt.1 | $3 \times 10^{(-01)}$ | $4 \times 10^{(-02)}$ | 7.44 | $3.8 \times 10^{(-13)}$*** |
| Rt.2 | $3.1 \times 10^{(-01)}$ | $4 \times 10^{(-02)}$ | 7.70 | $6.0 \times 10^{(-14)}$*** |
| Rt.3 | $2.4 \times 10^{(-01)}$ | 0.0401 | 6.04 | $2.7 \times 10^{(-09)}$*** |
| Ct.t | 5.79 | 1.04 | 5.56 | $4.0 \times 10^{(-08)}$*** |

Residual standard error: 67,030

| | |
|---|---|
| R-squared: 0.868 | Adjusted R-squared: 0.867 |
| F-statistic: 907.6 | P-value: $2.16 \times 10^{(-15)}$*** |

**Note:**
*** Significant at 1%.

significant at the 0.5% level $(P < 2.1 \times 10^{(-15)})$, the R-squared value is 82.94%. The fitted model can be written as:

$$x_t(Confirm) = 7.96 \times 10^2 + 8.41 \times 10^{-1}x_{t-1} - 2.21 \times 10^{-1}x_{t-2} \\ + 1.42 \times 10^{-1}x_{t-3} + 4.25 \times 10^{-3}w_{t-1} + \varepsilon_t \tag{6}$$

Table 4 shows the summary of the ARDL model, the confirmed cases recorded on the first day, second day and the third days. The daily recover cases of current, 1 day, 2 days and 3 days before have a significant impact on the number of daily recover cases from the COVID-19 on that particular day. The model is significant at the 0.5% level $(P < 2.16 \times 10^{(-16)})$ the R-squared value is 86.8%. We select the model using adjusted R-squared value and alpha value (*Benjamin et al., 2018*). The fitted model can be written as:

$$w_t(Recovered) = -8.1 \times 10^{(3)} + 5.79x_t + 3 \times 10^{(-1)}w_{t-1} \\ + 3.1 \times 10^{(-1)}w_{t-2} + 2.4 \times 10^{(-1)}w_{t-3} + \varepsilon_t \tag{7}$$

We evaluate models including RF, SVM, KNN, and ANN to compare their performance using various accuracy metrics including ME, RMSE, MAE, MPE and MAPE. These metrics provide different perspectives to assess predicting models. The first three are the absolute performance measures while the fourth and fifth are relative performance measures. The training sample is used to estimate the parameters for specific model architecture. The testing set is then used to select the best model among all models considered. Table 5 summarizes the RF, SVM, KNN, and ANN forecasting accuracy measures for the training set of COVID-19 daily deaths data.

In Table 5, the values of ME for RF, SVM, KNN, and ANN models reveal that RF shows the lowest value (the best) among the other methods. Similarly, the RMSE values of RF, SVM, KNN, and ANN, respectively show that the ANN achieved better performance compared to the other methods. Moreover, the MAE values indicate that ANN is better than the other methods. While the values of MPE, the RF achieved better performance compared to the other methods. Similarly, the values of MAPE and SMAPE revealed that

**Table 5 Forecasting accuracy measures of all models for daily deaths of training data.**

| Method | Error measurement tools | | | | | | |
|--------|------|-------|-------|-------|-------|-------|----------|
| | ME | RMSE | MAE | MPE | MAPE | SMAPE | R-square |
| RF | 0.156 | 4.54 | 0.937 | 0.021 | 11.54 | 7.67 | 0.870 |
| SVM | 0.511 | 10.01 | 2.70 | 0.018 | 10.23 | 8.98 | 0.849 |
| KNN | 0.953 | 17.82 | 9.71 | 0.581 | 16.65 | 10.23 | 0.770 |
| ANN | 0.166 | 0.027 | 0.002 | 0.002 | 8.21 | 6.38 | 0.912 |

**Table 6 Forecasting accuracy measures of all models for daily deaths of testing data.**

| Method | Error measurement tools | | | | | | |
|--------|---------|----------|----------|-------|-------|---------------------|----------|
| | ME | RMSE | MAE | MPE | MAPE | SMAPE | R-square |
| RF | −11.08 | 885.55 | 689.90 | −1.28 | 6.615 | 1.61 | 0.895 |
| SVM | 875.79 | 1,495.68 | 959.78 | 6.54 | 7.37 | 2.00 | 0.749 |
| KNN | −204.56 | 1,852.19 | 1,456.52 | −4.06 | 14.28 | 3.42 | 0.703 |
| ANN | 0.487 | 0.704 | 0.664 | 0.002 | 0.003 | $9 \times 10^{(-3)}$ | 0.881 |

ANN is less than one which indicates that the selected model falls in the range of perfect model (*Ahmadini et al., 2021*; *Gao et al., 2019*). Moreover, the R-square value of ANN is greater than other methods. We highlighted the results for the ANN model indicating the smallest value among all models. In most cases, the ANN method shows significant performance compares to the rest of the method's base on training parts.

In Table 6, the value of ME for the ANN model is lower than the other models. The results indicate that ANN shows the lowest value among the other methods. In addition, the ANN predicted value is close to the actual value. The ME value for KNN is negative and it reveals that the predicted value is less than the actual value. Similarly, the RMSE and MAE values of ANN are smaller than the rest of the methods and show that ANN achieved better performance compared to the other methods. Moreover, the MPE values are also smaller than the other methods. This shows ANN is better as compared to the other methods. While the values of MAPE and SMAPE of the ANN model are better than the three methods. Thus, the value of the MAPE and SMAPE for ANN is less than one which indicates that the selected model falls in the range of the perfect model (*Gao et al., 2019*). In addition, the value of R-square of all other methods are smaller than the ANN. This show that, ANN is better than the other methods. We highlighted the results for the ANN model indicating the smallest value among all models. The ANN method shows significant performance compares to the rest of the method's base on 20% testing parts in most of the cases. Figure 2 shows the plot of the forecasting accuracy measures for the models.

It is clear from the above plot that on average, ANN is the best model for forecasting the daily deaths of the COVID-19 outbreak. Table 7 summarizes the RF, SVM, KNN, and ANN forecasting accuracy measures of the COVID-19 confirm patient's on the training dataset.

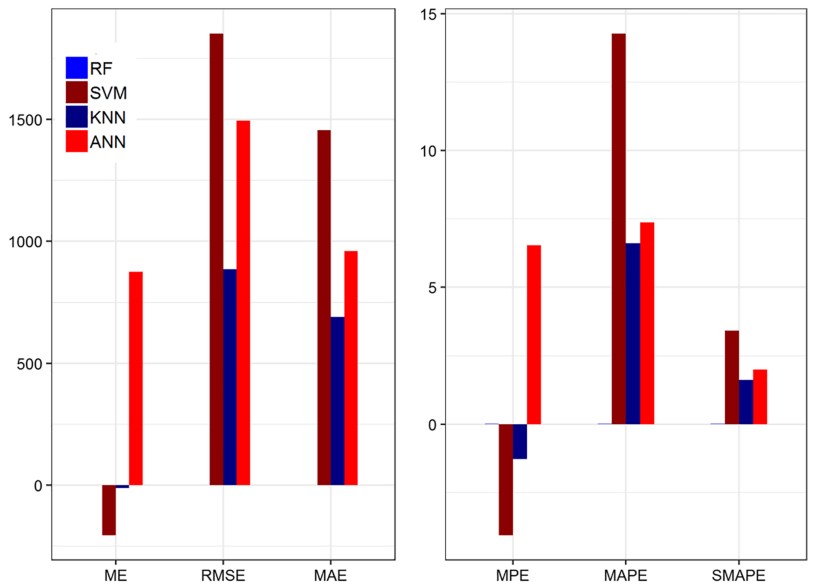

**Figure 2 Plot of the forecasting accuracy measures of different models for daily deaths.** The MAPE value is in the range of 0 to 1 which falls under the category of highly accurate forecasts (*Aamir, Shabri & Ishaq, 2018*; *Xu et al., 2020*).               

**Table 7 Forecasting accuracy measures of all models for daily confirm patients on training data.**

| Method | Error measurement tools | | | | | | |
|---|---|---|---|---|---|---|---|
| | ME | RMSE | MAE | MPE | MAPE | SMAPE | R-square |
| RF | 518.54 | 13,002.10 | 4,023.78 | −4.29 | 3.01 | 0.901 | 0.769 |
| SVM | 5,510.04 | 39,480.37 | 12,894.48 | −190.89 | 200.34 | 5.93 | 0.715 |
| KNN | 984.08 | 27,831.09 | 5,871.91 | −0.806 | 3.65 | 18.23 | 0.699 |
| ANN | −0.011 | 10.08 | 8.07 | −0.111 | 3.22 | 0.099 | 0.805 |

In Table 7, the value of ME of the ANN model is smaller than the rest of the methods. This indicates that the ANN predicted value is near to the actual value. KNN has the lowest (the best) value among the other methods with the highest accuracy. Similarly, the RMSE values of the ANN have shown the lowest RMSE value as compared to the rest of the methods. While the MAE and MPE values of the ANN model have the smallest value among the other methods. The values of MAPE and SMAPE of the ANN and KNN models are smaller than the other methods. Thus, the value of the MAPE for KNN is in the range of 1 to 10 which reveals that the selected model falls in the category very good model. In the same way, the R-square value of ANN is better than the other methods. Overall, the ANN method achieved significant performance better than the other methods based on training parts. This indicates that ANN results are more consistent with RF, SVM, and KNN.

In Table 8, the value of ME for the ANN model has the lowest (the best) value among the other methods with the highest accuracy. In the same way, the RMSE values and MAE values of the ANN model indicate that it predicted value close to the actual value.

**Table 8 Forecasting accuracy measures of all models for daily confirm patients on testing data.**

| Method | Error measurement tools | | | | | | |
|--------|------|-------|------|------|------|-------|----------|
| | ME | RMSE | MAE | MPE | MAPE | SMAPE | R-square |
| RF | 2,066.93 | 5,089.01 | 3,732.69 | 2.59 | 7.83 | 1.23 | 0.784 |
| SVM | 5,107.06 | 10,094.99 | 7,956.03 | 6.00 | 11.09 | 3.09 | 0.743 |
| KNN | 1,684.03 | 6,609.65 | 4,687.04 | 0.587 | 7.98 | 1.17 | 0.720 |
| ANN | −37.58 | 65.08 | 40.04 | −0.045 | 0.014 | 0.010 | 0.794 |

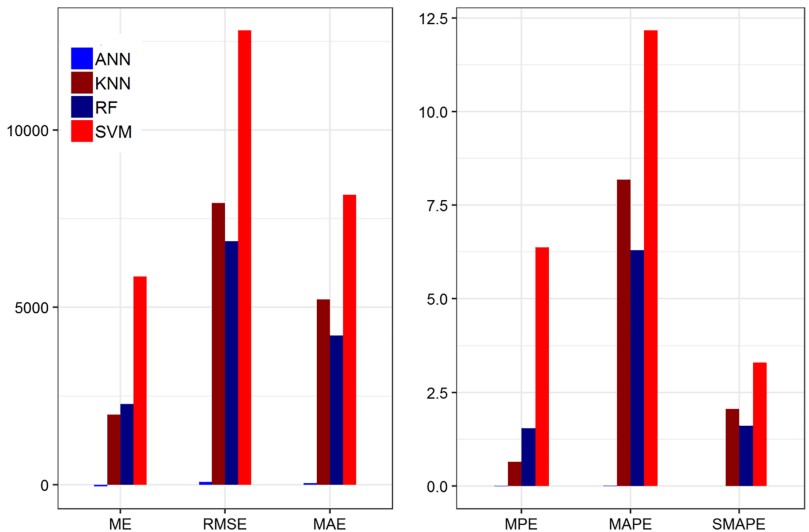

**Figure 3 Plot of the forecasting accuracy measures of different models for confirm cases.** The best model is ANN their ME = −37.58, RMSE = 65.08, MAE = 40.04, MPE = −0.045, MAPE = 0.014 and SMAPE = 0.010. The MAPE value is in the range of 0 to 1 which falls under the category of highly accurate forecasts.

The MPE value of the ANN model revealed that the ANN has the smallest value among the other methods. While the MAPE and SMAPE values tell that the ANN has the smallest value among the other methods, and it is in the range of 1 to 10 which revealed that the selected model falls in the category very good model. Furthermore, the R-square value indicate that, ANN is better than the other methods. On average, the ANN method achieved significant performance better than the other methods based on 20% testing parts. This indicates that ANN results are more consistent with RF, SVM, and KNN. Figure 3 shows the plot of the forecasting accuracy measures for different models.

In Table 9, the ME and RMSE values of the ANN model have the lowest (the best) value among the other methods with the highest accuracy and it reveals that the predicted value is very close and captured the original data. Similarly, the MAE and MPE values of the ANN model have the smallest value among the other methods and reveals that ANN has better power to capture the real data as compared to the other methods. Similarly, the values of MAPE and SMAPE of ANN and RF models are better than the other methods respectively. Thus, the value of the MAPE for ANN is in the range of 1 to 10 which showed

**Table 9 Forecasting accuracy measures of all models for daily recover patients on training data.**

| Method | Error measurement tools | | | | | | |
|--------|------|-------|-------|---------|--------|--------|----------|
| | **ME** | **RMSE** | **MAE** | **MPE** | **MAPE** | **SMAPE** | **R-square** |
| RF | 389.09 | 20,002.00 | 4,991.00 | −6.03 | 9.55 | **1.07** | 0.743 |
| SVM | 1,604.55 | 45,065.43 | 9,076.29 | −1,100.08 | 1,008.58 | 6.00 | 0.719 |
| KNN | 4,076.37 | 42,009.98 | 13,060.40 | −8.09 | 17.13 | 20.32 | 0.765 |
| ANN | −0.003 | 4.08 | 4.09 | −0.243 | 0.532 | 1.78 | 0.784 |

**Table 10 Forecasting accuracy measures of all models for daily recover patients on testing data.**

| Method | Error measurement tools | | | | | | |
|--------|------|-------|-------|---------|--------|--------|----------|
| | **ME** | **RMSE** | **MAE** | **MPE** | **MAPE** | **SMAPE** | **R-square** |
| RF | 25,809.89 | 6,497.09 | 3,703.80 | 4.60 | 7.96 | 1.84 | 0.773 |
| SVM | 10,006.09 | 15,120.26 | 10,089.76 | 15.82 | 15.52 | 4.92 | 0.714 |
| KNN | 9,021.67 | 12,650.43 | 9,489.76 | 17.05 | 18.89 | 6.09 | 0.767 |
| ANN | −664.04 | 1,057.52 | 692.36 | −0.108 | 0.108 | 0.053 | 0.817 |

that the selected model falls in the category very good model. In addition, the value of R-square reveals that the other methods are less efficient than ANN. On average, the ANN method achieved significant performance better than the other methods based on 20% testing parts. This indicates that ANN results are more consistent with RF, SVM, and KNN. Figure 3 shows the plot of the forecasting accuracy measures for different models.

In Table 10, the ME and RMSE values for the ANN model have the lowest value among the other methods with the highest accuracy. The MAE value and MPE value indicate that ANN has the smallest value among the other methods. Moreover, ANN follows the real data pattern with the smallest error as compared to the other methods. Similarly, the values of MAPE and SMAPE for ANN are in the range of 1 to 10 which revealed that the selected model falls in the category very good model. While the R-square value of ANN is higher than the other methods. On average, the ANN method achieved significant performance better than the other methods based on 20% testing parts. This indicates that ANN results are more consistent with RF, SVM, and KNN. Figure 4 shows the plot of the forecasting accuracy measures for different models.

## DISCUSSION

The performance of the neural network model can be assessed once trained the network employing the performance function as a prediction. All the methods are capable of capturing the pattern of the data effectively. Moreover, ANN performed well and almost capture the whole pattern of the testing part of the data when compared to RF, SVM, and KNN methods. Figure 3 shows the prediction accuracy of the number of daily Covid-19 recovered cases of RF, SVM, KNN, and ANN methods. The world daily deaths

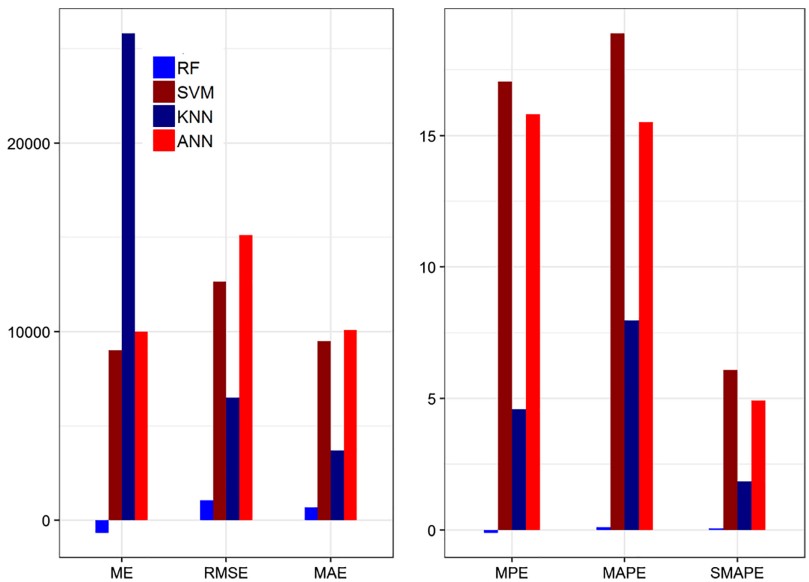

**Figure 4** **The plot of the forecasting accuracy measures of different models for recover cases.** The best model is ANN their ME = −664.04, RMSE = 1,057.52, MAE = 692.36, MPE = −0.108, MAPE = 0.108 and SMAPE = 0.053.                                   

original testing data of COVID-19 and the forecasted data for RF, SVM, KNN and ANN models are plotted in Figure 5.

Figure 5 displays the prediction accuracy of RF, SVM, KNN, and ANN models. All the models are capable of capturing competently the pattern of the daily death cases of COVID-19. Figure 5 clearly shows that ANN captured the pattern of the test set of the data better than RF, SVM, and KNN methods. Also, Figure 5 displays the prediction accuracy of RF, SVM, KNN, and ANN models for COVID-19's daily recovered cases. Similar to death cases accuracy results, all the models effectively captured the pattern of the daily recovered cases of COVID-19. In the same way, in Figs. 6 and 7, the ANN captured the pattern on the test part of the data. While the rest of the methods first follow the pattern up to some extent and then insensitive to the original data. Figures 6 and 7 are shown below.

In Fig. 8, the original COVID-19 number of deaths data points and the resulting forecast of ANN were plotted for the next 15 days from (19 August 2021 to 2 September 2021). As shown in the figure, the ANN forecast captures and follows the pattern of the original death cases of COVID-19. The subsequent 15 days forecasted line fluctuated near 10,000. In addition, the forecasted number of deaths tends to gradually upward over time. This is an indication that the number of daily deaths increases over time.

In Fig. 9, the original COVID-19 confirmed patient's data and forecast of ANN exhibited for the next 15 days from (19 August 2021 to 2 September 2021). The ANN model forecast captured the pattern of the original COVID-19 confirms patient data. In addition, the next 15 days forecasted drift going in the upward direction. This reveals that the number of daily confirm is increasing over time.

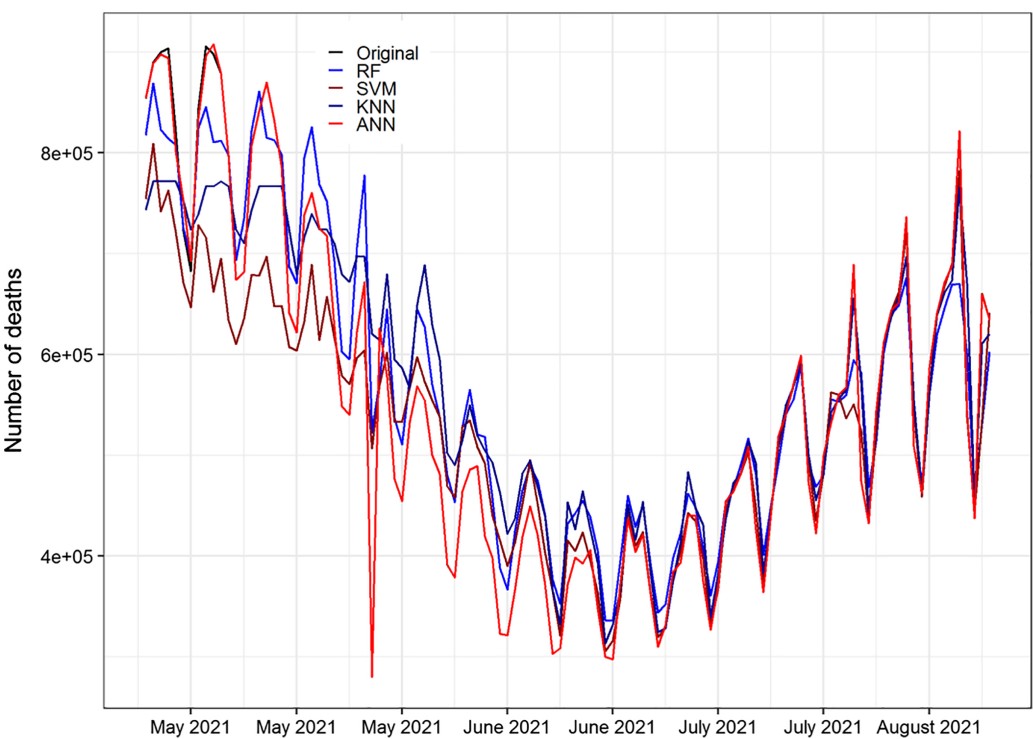

**Figure 5 Original and forecasted values of RF, SVM, KNN, and ANN models for daily death cases of COVID-19 of the testing set.** The testing set consist of 20% of the daily deaths a total of 115 observations spanning from 25 April 2021, to 18 August 2021 to validate.

In Fig. 10, the original COVID-19 recovered patient's data and forecast of ANN exhibited for the next 15 days from (19 August 2021 to 2 September 2021). The ANN model forecast captured the pattern of the original COVID-19 recover patient's data. In addition, the next 15 days forecasted drift going in the upward direction. This reveals that the number of daily recoveries is decreasing over time.

The key findings of this work as follows:

- The machine learning approaches are compared in this study to predict the Covid -19 cases.
- The ANN results on average are better than the other methods using the performance metrics and used to forecast the next 15 days' values.
- The forecast shows that in the next 15 days the total number of death cases will increase using ANN.
- The confirmed cases forecast for the next 15 days revealed that the number of recovered cases will increase.
- The recovered cases forecast for the next 15 days revealed that the number of recovered cases will decrease.

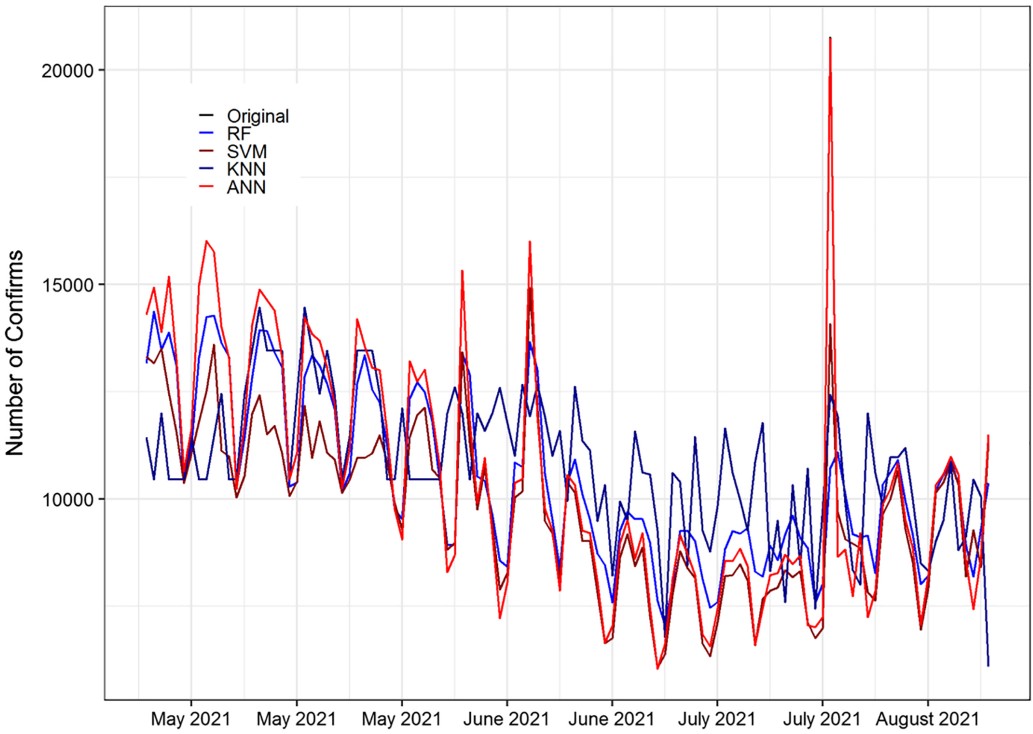

**Figure 6 Original and forecasted values of RF, SVM, KNN, and ANN models for confirmed cases of COVID-19 of the testing set.** The testing set consist of 20% of the daily confirms a total of 115 observations spanning from 25 April 2021 to 18 August 2021 to validate.

- From this study, it is revealed that the ANN provides the best forecast for the short term. Therefore, policymakers can use this technique to take up-to-date decisions for the short-term plan.

## Limitations

- In this study, we do not consider and measured the other parameters like the number of lockdowns, social distancing, and measure of self-isolation.
- The current study did not measure the association of vaccinated people and the number of new daily cases.

## Future work

In this study, the RF, SVM, KNN and ANN algorithms are used, though all the algorithms captured the original track almost in all cases, *i.e.* for the daily confirmed cases, deaths, and the number of recovered cases of the four countries. However, the performance metrics suggested the ANN model. Moreover, it can be possible to consider the other parameters like the number of lockdowns in the country, the number of vaccinations to the people, treatment procedures, *etc.* that can help for government to make and adjust their policies according to the various cases that are forecasted.

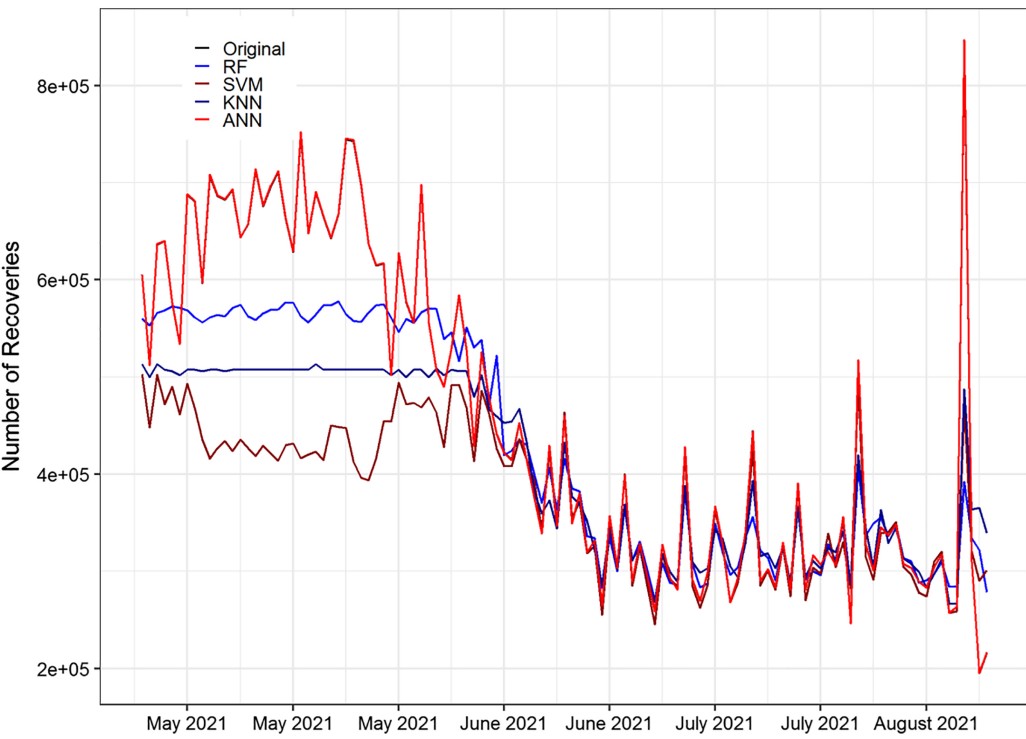

**Figure 7 Original and forecasted values of RF, SVM, KNN, and ANN models for recover cases of COVID-19 of the testing set.** The testing set consists of 20% of the daily recovered cases and a total of 115 observations spanning from 25 April 2021 to 18 August 2021.

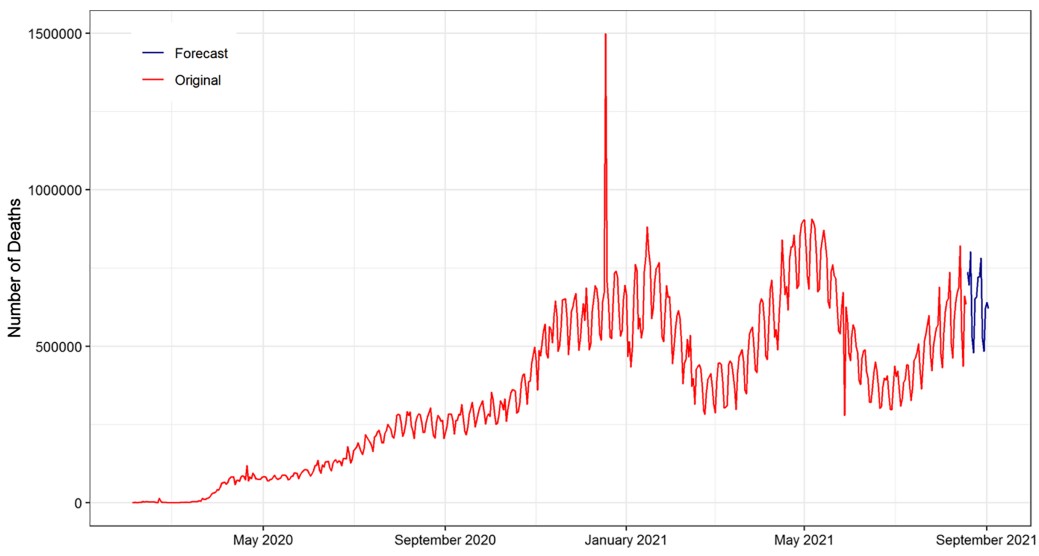

**Figure 8 Plot of the original and forecasted values of ANN model for daily deaths cases of COVID-19.** The blue line shows the 15 days ahead forecasts spanning from 19 August 2021 to 02 September 2021. The forecasted number of deaths tend to gradually increasing over time.

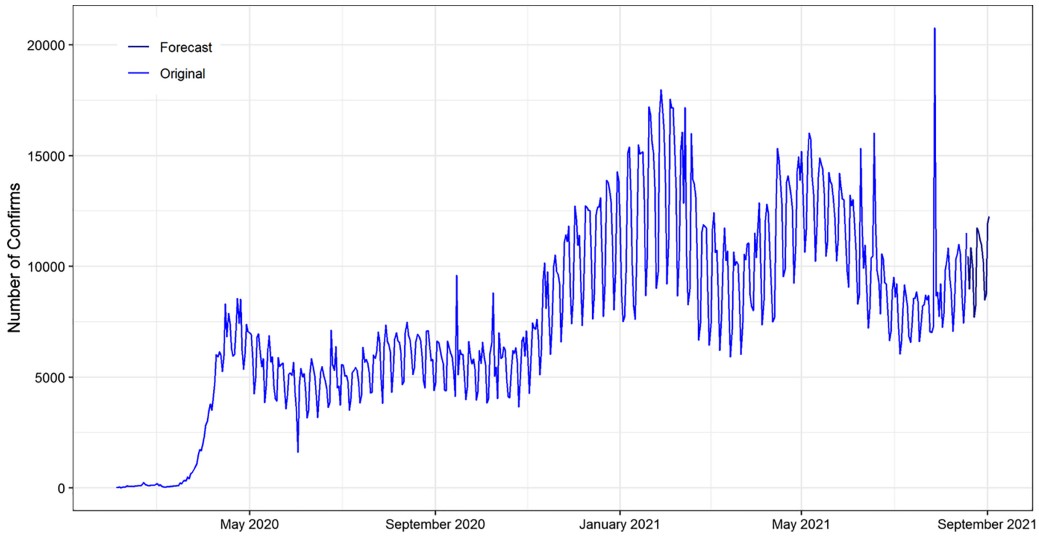

**Figure 9 Plot of the original and forecasted values of ANN model for daily confirm cases of COVID-19.** The Navy Blue line show the 15 days ahead forecasts spanning from 19 August 2021 to 02 September 2021. The forecasted drift going in the upward direction over time.

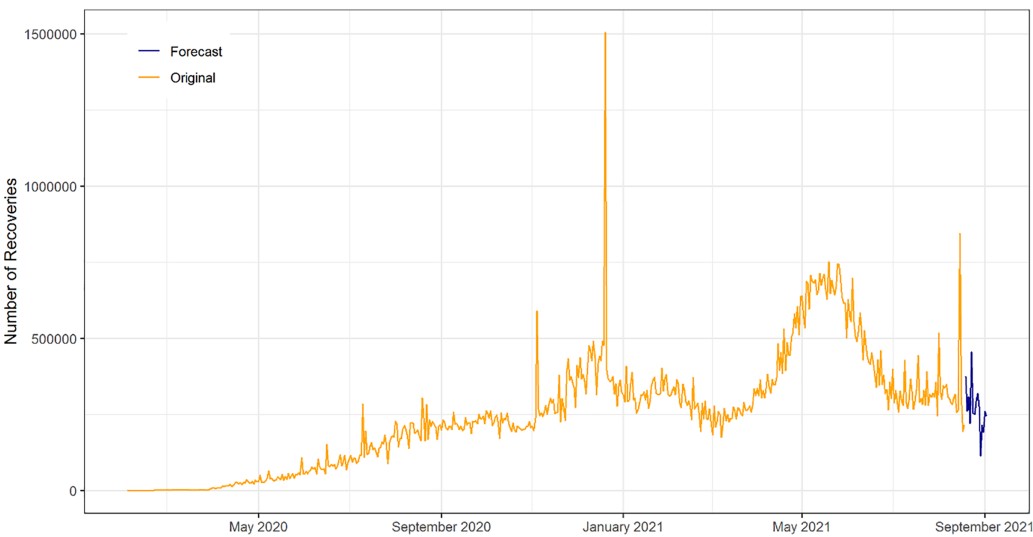

**Figure 10 Plot of original and forecasted values of ANN model for daily recovered cases of COVID-19.** The blue line show the 15 days ahead forecasts spanning 19 August 2021 to 02 September 2021. The forecasted drift going in downward direction.

## CONCLUSIONS

This paper proposed four predicting models for the COVID-19 outbreak. The methods are compared with respect to six performance metrics including ME, RMSE, MAE, MPE, MAPE, and SMAPE. The results for the daily deaths cases are based on 80% training and 20% testing parts. Among the four methods using these performance metrics, the ANN achieved better results in every aspect. In the same way, the results obtained for the daily

recovered cases using 80% training and 20% testing parts and ANN have attained better results than the other methods. Moreover, daily confirm cases results obtained using the same training and testing parts and in most of the cases, ANN performed better than the other methods. Therefore, the major findings of this study reveal that ANNs outperform the rest of the methods for both models. In addition, ANN suggests consistent prediction performance compared to RF, SVM, and KNN models and hence preferable as a robust forecast model. The AI-based method's accuracy for predicting the trajectory of the COVID-19 is high. For this specific application in predicting the disease, the authors consider the results are reliable. In this study, ANN generates the fastest convergence and good forecast ability in most cases. The AI results can help in short-term plans for the disease occurrences. The estimate models will help the public authority and medical staff to be prepared for the coming situation and take further timelines in medical care structures. The forecasted figures were calculated for the next 15 days (19 August 2021 to 2 September 2021) for COVID-19 data. Predicting an event is a difficult, and some customized models probably would not be generalized to the cultural and financial conditions of various countries. In this study, the proposed models do not considers the factors like area and other government strategies. Therefore, it is to be noted, while take to mean these predictions.

### Funding
The authors received no specific funding for this work.

### Competing Interests
The authors declare that they have no competing interests to disclose.

### Author Contributions
- Muhammad Naeem conceived and designed the experiments, performed the experiments, performed the computation work, prepared figures and/or tables, and approved the final draft.
- Jian Yu conceived and designed the experiments, analyzed the data, performed the computation work, authored or reviewed drafts of the paper, and approved the final draft.
- Muhammad Aamir conceived and designed the experiments, performed the experiments, performed the computation work, prepared figures and/or tables, and approved the final draft.
- Sajjad Ahmad Khan analyzed the data, authored or reviewed drafts of the paper, and approved the final draft.
- Olayinka Adeleye performed the experiments, analyzed the data, authored or reviewed drafts of the paper, and approved the final draft.
- Zardad Khan analyzed the data, authored or reviewed drafts of the paper, and approved the final draft.

## Data Availability

The data are available in the Supplemental Files and GitHub: https://github.com/CSSEGISandData/COVID-19.

## Supplemental Information

Supplemental information for this article can be found online at http://dx.doi.org/10.7717/peerj-cs.746#supplemental-information.

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
