# Peer review of "Comparative analysis of machine learning approaches to analyze and predict the COVID-19 outbreak"

_PeerJ Computer Science, doi:10.7717/peerj-cs.746_

## Round 0.1 · original submission · Major Revisions

The article presents a study based on machine learning and statistical methods employed to predict the COVID-19 outbreak.

The article is interesting but, as the reviewers pointed out, it needs some improvement.

In particular, the authors should add some justification regarding the methods employed and add a literature review to contextualize the study in the current scientific literature.

Here are some aspects to address.

Major points:

1- The results should be measured with the coefficient of determination R-squared and SMAPE, in addition to ME, RMSE, MAE, MPE, and MAPE. The authors should give more importance to the R-squared results in the article.

2- All the information about the software packages employed by the authors should be moved into a separate ad-hoc section.

3- The statistical significance in Table 2 and in the text should be reported for p-values lower than the 0.005 threshold, as pointed out by Benjamin and colleagues ( https://doi.org/10.1038/s41562-017-0189-z ).

4- The values with exponential numbers should be written in a scientific notation. For example, "8.97E-05" should be rewritten as "8.97 x 10^(-5)".

Minor points:

5- Please write the date months as complete words ("Jan" --> "January", "Feb" --> "February", etc).

6- All the occurrences of "i.e." should be replaced with "that is" or "that means".

7- The style of the text must be improved. Terms like "In (Fig. 10)" should be replaced with "In Fig. 10".

8- The values of Tables 6, 7, 8, and 9 should be aligned on the left.

9- In the figures, the axes values should be written in horizontal position, not in diagonal position.

10- In the figures, the y axis label should say "number of" instead of "No of".

Please also consider and address the comments of the two reviewers.

Reviewer 1 ·

Basic reporting

The current study does not explain why these models were selected for comparison. Specifically, it lacks a review or summary of the existing studies that have applied forecasting models for the COVID-19 pandemic. Please see the existing papers in PubMed/LitCovid (https://www.ncbi.nlm.nih.gov/research/coronavirus/docsum?text=forecasting%20model). The methods are from traditional mathematical models and recurrent neural networks other than ANN. The study should discuss them in detail and select representative models with reasonings.

Experimental design

First, the study should explain the data split and cross-validation in more detail. How was the cross-validation performed? Since this is time-series data, the split by time is critical. How does the 10-fold cross-validation reflect the timeline?

Second, a more religious design should be made for the forecasting model. Currently, the spanning between the train and testing is too close. The training set contains the data up to 7th Nov 2020, whereas the testing data is directly from 8th Nov 2020. Arguably, if the data of 7th Nov 2020 is available, it is not too challenging to predict the 8th even using a simple model. And it is not very useful because there is not much time for interventions or prevention can be done in advance. The study could quantify how the model forecasts in a week or earlier in advance.

Validity of the findings

The study lacks a thorough discussion on the limitations. For example, there are many factors (prevention, vaccinations etc) that could impact the case growth. Such factors cannot be captured in the models.

Reviewer 2 ·

Basic reporting

The manuscript size should be reduced, by improving the writing style. For example, information included in Tables, should not be also mentioned in the main body of the manuscript in such detail. E.g., lines 269-281, lines 283-296, lines 301-313, etc.

Replace these details with a more qualitative and descriptive discussion of the overall findings.

The literature review on the topic is very poor at the moment, and should be enhanced.

The connection of this work with previously published work in the Journal should be established.

The manuscript needs very thorough editing and proofreading.

It would be best to remove the gray background of the figures.

The green in Figure 2 is too bright; replace it with a smoother one (use the same blue and red as in Figure 1).

Experimental design

The research question is well defined, and it contributes to the international literature on the topic. The methods are adequately described.

Validity of the findings

This work has merit to be an important contribution to the topic of using machine learning methods for COVID-19 forecasting, by comparing several such methods.

Additional comments

This is an interesting manuscript that has a contribution to the international literature of COVID-19 forecasting, by comparing machine learning approaches towards this direction. There are some issues that need to be addressed before the manuscript is ready for publication.

---

## Round 0.2 · Minor Revisions

The authors improved the article substantially but still need to address two points:

1) The regression analysis results should be measured with R-squared (coefficient of determination), as indicated in the previous review report. R-squared is the most informative and truthtful metric to measure regression analyses.

2) As indicated by the reviewer, it would be much better if the authors tested their machine learning models on an external dataset. Did the authors look for one? Did they find something? Please specify.

Reviewer 1 ·

Basic reporting

The reviewers have addressed my concerns.

Experimental design

Thanks, reviewers for the efforts in addressing my comment #3. However, this comment has not been fully addressed. The statement 'Actually dividing the data into training and testing parts is used to evaluate the model performance. The model that are trained and the data are tested is completely unseen to the model. ' made by the author is fine but is not enough for the evaluation on generalization. Splitting training and testing from the same dataset of course is a way to evaluate the performance. However, splitting from the same dataset will make the training and testing set share similar distributions or characteristics, which is not always the case when applying to new data. In this specific case of outbreak prediction, it is critical to apply to an external dataset that does not have overlapped periods to the training and testing datasets.

Validity of the findings

I do not have additional comments.

---

## Round 0.3 · Minor Revisions

The authors addressed most of the previous comments and requests, but they forgot to include the results measured with R-squared.

Please add the formula of R-squared in Table 1, and its measured results in Tables 5, 6, 7, 8, 9, and 10.

Reviewer 1 ·

Basic reporting

My last concern has primarily been addressed. Thanks authors for their dedicated efforts

Experimental design

N/A

Validity of the findings

N/A

---

## Round 0.4 · accepted · Accept

The authors addressed correctly all the requested changes and therefore I recommend this article for publication.